# Green Roofs Affect the Floral Abundance and Phenology of Four Flowering Plant Species in the Western United States

**Kyle Michael Ruszkowski [1],*** and **Jennifer McGuire Bousselot [2],***

1   Graduate Degree Program in Ecology, Colorado State University, Fort Collins, CO 80523, USA
2   Department of Horticulture and Landscape Architecture, Colorado State University,
    Fort Collins, CO 80523, USA
*   Correspondence: k.ruszkowski@colostate.edu (K.M.R.); jennifer.bousselot@colostate.edu (J.M.B.)

**Abstract:** This study investigates the potential for green roofs to support pollinator diversity and abundance in urban ecosystems through the altered floral phenology and floral abundance of plants. Floral phenology and the floral abundance of green roof plants are compared to plants grown at grade on the Front Range in Fort Collins, Colorado, and how these changes may affect pollinator biodiversity in urban ecosystems. An independent block design is employed, within one green roof and one ground-level garden, approximately 120 m apart, with replicate plants of 4 species in each garden. Pollinator observations were made weekly during the bloom period for each species. Blue vane traps were used to passively measure pollinator diversity along a transect between the green roof sites and the sites at grade. The total number of flowers per plant is variable between site types, depending on the plant species. However, all species of plants tested bloomed earlier when grown on the green roof than when grown at grade. Pollinator abundance and diversity were low at both site locations. Green roofs may provide foraging opportunities earlier in the season in temperate regions, which can extend the duration of floral foraging opportunities when supported by green infrastructure at grade.

**Keywords:** floral resources; green roof; phenology; pollinators

## 1. Introduction

Urban ecosystems rely on pollination to produce fruits, vegetables, and seeds for both human and wildlife consumption. However, urbanization alters plant–pollinator interactions in ways that may be detrimental [1,2]. North America, and the rest of the globe, is experiencing an unprecedented insect decline across many taxa [3]. This loss of abundance and diversity has far-reaching implications for food systems and ecological systems. One particularly diverse and important group of pollinators is bees. There are approximately 20,000 unique bee species worldwide. It is estimated that Colorado's diverse landscape is home to 946 known native bee species across 66 genera [4]. Green infrastructure technologies, such as green roofs, may play an important role in protecting pollinator diversity and abundance in urban areas from habitat destruction and degradation.

Pollinator decline is caused by a confluence of known and unknown factors. Pesticide use in ornamental and food crop production combined with climate change are likely major factors negatively impacting pollinators globally [3]. Warming temperatures alter both the pollinator lifecycle and the lifecycle of the plants on which they rely [5]. Additionally, changes in land cover have become an obstacle for pollinators. Native landscapes across the globe have been converted into farmland and urban and peri-urban systems, which fragment and degrade the remaining native habitats. Currently, over 55% of the world's population lives in cities and that percentage is expected to grow to 68% by the year 2050 [6]. Approximately 90% of the US population is projected to live in urban settings by the year 2050 [6]. However, with the implementation of green infrastructure, the ecological burden

of land conversion to urban and suburban systems may be lessened and provide a path for people to exist in concert with a variety of plants and pollinators that are native to the region.

Green urban infrastructure inhabits a variety of forms, typically at grade (ground level). Green infrastructure implemented during city planning is most often designed to address stormwater challenges or to provide green amenity space for residents [7]. In addition to stormwater solutions, green infrastructures, such as parks, bioswales, green roofs, rain gardens, and even turfgrass, provide direct and indirect benefits to humans. These benefits include noise reduction, urban heat island mitigation, recreational space, and much more [8]. Habitat provisioning is another benefit provided by green urban infrastructure, but the efficacy of that habitat is dependent on the quality, density, and patch size of the green infrastructure [8,9].

Green roofs provide habitat for a variety of fauna, including birds and arthropods, by providing an urban environment necessary for nesting, resting, feeding, and breeding. It has been anecdotally noted by green roofing practitioners that green roof soilless substrates often reach higher diurnal temperatures than traditional garden analogs. Green roofs will generally have higher sun and wind exposure and lower water-holding capacity than ground-level gardens. Due to the unique growing environment on green roofs, the flowering phenology of some flowering plant species is likely to be altered. Because of these warmer, drier conditions on the green roof, some species of plants are likely to flower earlier in the season [10,11]. The warmer root zone temperatures may play a role in the mobilization of stored carbohydrates from the roots into the shoots for growth, flowering, and seed production [12,13]. Green roofs provide unique habitat and foraging opportunities for urban pollinators that would otherwise be unavailable, but more research on the effects of the green roof environment is needed to manage pollinator health and biodiversity more accurately in urban ecosystems.

Green roofs provide unique habitat opportunities for flora and fauna in urban ecosystems [14] that are not available on other roofing non-vegetated options. The earlier bloom times have potential synergies with urban pollinator conservation. Some bee species have altered phenology, emerging earlier in the season, likely related to global temperature increases [5]. Hypothesizing that the anticipated higher diurnal temperatures of green roofs compared to garden analogs at grade will modify the floral phenology of certain flowering plant species [15], green roofs may offer expanded foraging opportunities for pollinators that are not typically offered synchronously in analogous garden settings at grade.

## 2. Materials and Methods

This experimental study was conducted on the Front Range of the Rocky Mountains in Fort Collins, Colorado, on the campus of Colorado State University. Fort Collins, Colorado, is nestled between the Great Plains and the Front Range of the Rocky Mountains and experiences a temperate climate with warm summers and cold winters with about 500 mm of annual precipitation. The study site is in an urban environment and immediately surrounded by intensely managed green space and an urbanized landscape.

The green roof used in this study is located above the second floor of the Nutrien Agricultural Sciences Building (40.5732, −105.0808) on the campus of Colorado State University. The green roof's construction was completed in March 2022. The green roof is approximately 65 m$^2$. It is an intensive-style green roof with tapered depth. The depth ranges from approximately 30 cm in the shallowest section to 570 cm in the deepest section. The growing substrate is composed primarily of 60% expanded shale, 20% high-quality compost, 10% vermiculite, and 10% peat moss by volume. No additional substrate amendments were made during the study.

The ground-level study site (40.5730, −105.0825) is located approximately 137 m west of the Nutrien Agricultural Science Building's green roof. In March 2022, turfgrass was removed from the location to prepare the site for the research plants. Additionally, the native soils, which were high in clay content and had low drainage, were amended with

perlite and compost before planting. A 1.5 cm layer of compost and substrate was applied to the plot and then incorporated into the upper 12 cm of the soil surface.

Several plant species, both native and non-native to the Colorado Front Range, were selected for this study (Table 1). Plant selection was evaluated on multiple criteria. Plant species needed to be able to tolerate the different growing conditions of the green roof and at-grade sites. Because both study sites were constructed in the spring of 2022, we selected species that we expected to bloom in the same growing season that they were planted. Additionally, we selected species with corollas that were conducive to nectar sampling. Finally, we were limited by what species were commercially available. Asclepias incarnata, Ipomopsis aggregata, and Oenothera speciosa were purchased from the High Plains Environmental Center (Loveland, CO, USA). *Allium 'Millennium'* was purchased from Arbor Valley (Fort Collins, CO, USA). All plants in this study on the green roof and at grade were planted on 15 April 2022, after they were hardened off for seven days. Six other species were planted on the green roof and at grade but did not bloom during the study period. These species that did not bloom during the study were *Aquilegia chrysantha*, *Monarda fistulosa*, *Penstemon strictus*, *Penstemon barbatus*, *Salvia canescens* var. daghestanica, and *Sedum lanceolatum*.

**Table 1.** Plant common names, scientific names, and family names of the species used in this study.

| Common Name | Scientific Name | Family |
| --- | --- | --- |
| Millennium Chives | *Allium 'Millennium'* | Amaryllidaceae |
| Swamp Milkweed | *Asclepias incarnata* | Apocynaceae |
| Scarlet Gilia | *Ipomopsis aggregata* | Polemoniaceae |
| Evening Primrose | *Oenothera speciosa* | Onagraceae |

An independently replicated, blocked design was used in this experiment. Blocks 1, 2, and 3 were located on the green roof and blocked by depth. The shallowest point in Block 1 was 380 cm, and its deepest point was 455 cm, averaging 417.5 cm across the block. The shallowest point in Block 2 was 495 cm, and its deepest point was 540 cm, averaging 517.5 cm across the block. The shallowest point in Block 3 was 555 cm, and its deepest point was 565 cm, averaging 560 cm across the block. Blocks 4, 5, and 6 were in the site at grade and blocked by supplemental irrigation. Block 4 received a 15 min supplemental irrigation event every Monday, and Block 5 received a 5 min supplemental irrigation event every Monday. Block 6 did not receive any additional irrigation beyond what was provided by the overhead irrigation system. There was 30 cm of distance between each of the three blocks in the green roof site and in the site at grade. Each block was 3 m long by 1.5 m wide. Plant species replicates were randomly arranged in each row of a 5 × 10 array so that there were 5 replicates per plant species per block for a total of 15 replicates per species on the green roof and 15 replicates per species at grade. All plots were irrigated by an overhead irrigation system. Irrigation ran for 10 min on Mondays, Wednesdays, and Fridays. The irrigation system ran from 10 May 2022 to 1 September 2022.

Substrate volumetric water content, substrate temperature, and solar radiation were measured at the green roof and grade sites using HOBO weather stations (H21-USB, Onset Computer Corporation; Bourne, MA, USA). The substrate volumetric water content was measured for each block using a HOBO moisture sensor (S-SMC-M005, Onset Computer Corporation; Bourne, MA, USA) that was buried 11 cm below the surface of the substrate in the center of each block. The substrate temperature was measured for each block using a HOBO temperature sensor (S-TMB-M006, Onset Computer Corporation; Bourne, MA, USA) that was buried 11 cm below the surface of the substrate in the center of each block. On the green roof, one solar radiation sensor (A-LIB-M003, Onset Computer Corporation; Bourne, MA, USA) was placed equidistantly between Block 1 and Block 2, and the second sensor was placed equidistantly between Block 2 and Block 3. At the site at grade, one solar radiation sensor was placed equidistantly between Blocks 4 and 5, and the second solar radiation sensor was placed equidistantly between Blocks 5 and 6.

Flower count surveys were conducted every week during bloom time. Flower counts began during the week of the first bloom for a species and ended either when the species produced no new flowers for the season or when the irrigation was stopped on 1 September 2022. Flowers were only counted if the reproductive structures were intact. The same flower may have been counted over multiple weeks. For *Allium 'Millennium'* and *Asclepias incarnata,* the number of umbels that contained flowers with intact reproductive structures was counted instead of individual flowers. Flower count data could not be collected over the entire bloom time for *Oenothera speciosa* and *Ascleppias incarnata* because aphids and Japanese beetles destroyed many of the buds before flower development. After Week 10, only the presence or absence of flowers on *Oenothera speciosa* replicates could be determined. For *Asclepias incarnata*, only the presence or absence of flowers was recorded for the entirety of the growing season.

To assess plant health, height and width measurements were recorded once in Week 6 and again in Week 19 to create a plant size index for each of the replicates. Height measurements were taken from the surface of the substrate to the tallest apical meristem. In the case of *Allium 'Millennium'*, height measurements were taken from the substrate surface to the apex of the tallest leaf. The first width measurement was taken at the widest section of the plant, 10 cm above the surface of the substrate. The second width measurement was taken perpendicular to the first, widest measurement. A plant size index was calculated by averaging the values for both widths and heights.

At the end of the growing season when a plant stopped flowering (19 September 2022 for *Allium 'Millennium'* and *Oenothera speciosa,* 7 October 2023 for *Ipomopsis aggregata*, 12 October 2022 for *Asclepias incarnata*), the aboveground biomass was harvested, and replicates were individually bagged in labeled brown paper bags, dried, and weighed for each plant species. *Allium 'Millennium'*, *Asclepias incarnata*, and *Ipomopsis aggregata* were dried in a drying oven for 48 h at 105 degrees Celsius. *Oenothera speciosa* was dried at 70 degrees Celsius for 72 h.

The plant relative chlorophyll concentration measurements were made for *Allium 'Millennium'*, *Asclepias incarnata*, and *Oenothera speciosa* using an atLEAF CHL Plus Handheld Chlorophyll Meter (FT Green LLC, Wilmington, DE, USA). Chlorophyll content was not measured for Ipomopsis aggregata due to the shape of the chlorophyll meter and the highly dissected, lanceolate structure of the I. aggregata leaf. All measurements were made between 9:00 am and 11:00 am on sunny mornings to avoid the effects of the time of day and the angle of the sun on the measurements. To take the measurement, the leaf apex was placed into the handheld meter and the leaf midrib was adjusted to be in the center of the light sensor.

Following the protocol developed by Mason, Kondratieff, and Seshadri [16], each plant replicate was observed for two minutes between 9:00 am and 11:00 am once every week during their flowering period. Pollinators were identified to morphospecies during the in situ observations. Morphospecies groups were created using easily identifiable visual cues and are comprised of the most common genera of bees found in Colorado. The morphospecies groups are honey bees (*Apis mellifera*), hairy leg bees (*Anthophera* sp., *Diadasia* sp., *Melissodes* sp., and *Svastra* sp.), hairy belly bees (*Anthidium* sp., *Megachile* sp., *Osmia* sp., and *Hoplitis* sp.), bumble bees (*Bombus huntii*, *Bombus centralis*, *Bombus griseocollis*, *Bombus morrisoni*, *Bombus nevadensis*, *Bombus occidentalis*, *Bombus insularis*, and *Bombus fervidus*), tiny dark bees (*Ceratina neomexicanum*, *Ceratina* sp., *Hylaeus* sp., and *Lasioglossum* sp.), striped sweat bees (*Halictus* sp. and *Lasioglossum* sp.), and cuckoo bees (*Nomada* sp., *Sphecodes* sp., and *Epeolus* sp.). Visiting pollinators were counted if they contacted floral reproductive structures during the two-minute observational window [16]. A plant species replicate was only observed if it had at least one flower with intact floral reproductive structures.

Five blue vane traps (BanfieldBio Inc., Seattle, WA, USA) were set up along a transect from the plots at grade to the green roof, with one trap in the center of the plots at grade, two along the transect, one in the center of the green roof, and on the non-green roof of the

Plant Science Building at Colorado State University in Fort Collins, Colorado. The trap on the non-green roof was approximately 13 m north of the plots at grade.

The traps were placed in their respective locations from 8 June 2022 to 10 June 2022 and 18 July 2022 to 20 July 2022 from 9:00 am to 3:30 pm each day. Samples were collected from the trap at the end of each sampling day and stored in vials filled with 70% isopropyl alcohol. The blue vane traps were filled with a one-to-ten ratio of Dawn Dish Soap to water. Samples collected from the traps were identified to morphospecies groups described by Mason et al. (2018).

All statistical analyses were conducted in RStudio, version 1.41717 (RStudio, Inc., Boston, MA, USA). Using blocks as independent replicates, a comparison between the green roof sites and sites at grade was conducted. Flower count data were analyzed using a general linear mixed-effects model with site-by-time interactions as the fixed effects and block as the random effect, and a Poisson Distribution was assumed to model the count data [17]. The week was calculated as a categorical variable in this model.

A linear mixed-effects model was used to analyze the aboveground biomass and relative chlorophyll content. Each block was considered an independent replicate and used to make a direct comparison between the green roof sites and sites at grade. Site location (green roof or at grade), week, and site-by-week interactions were the fixed effects in the model, and the block and replicate within the block were the random effects of the model.

## 3. Results and Discussion

The green roof environmental conditions were generally more extreme than the conditions at grade. The minimum temperatures for the plots on the green roof and the plots at grade showed the smallest difference between the two site types. In every month other than July, the minimum temperatures on the green roof were greater than those at grade (Table 2). In July, the minimum temperature recorded was the same for both the green roof and the site at grade. The green roof maximum recorded temperatures were higher than at grade in every month. June, July, and August showed the highest differences in maximum temperature between the green roof plots and the plots at grade. The green roof had higher average temperatures across all months of the growing season.

**Table 2.** The minimum, maximum, and mean monthly temperatures that were recorded on the green roof and at grade.

| Month | Site Location | Min. Temp. (°C) | Max. Temp. (°C) | Mean Temp. (°C) |
|---|---|---|---|---|
| May | Roof | 13 | 16.8 | 20.3 |
| | Ground | 11.1 | 14.2 | 14 |
| June | Roof | 18 | 37.3 | 27.5 |
| | Ground | 14.7 | 21 | 18.5 |
| July | Roof | 18.5 | 37.5 | 30 |
| | Ground | 18.5 | 23 | 25.8 |
| August | Roof | 23.9 | 40.3 | 31.3 |
| | Ground | 18.2 | 22.2 | 22.5 |
| September | Roof | 16.4 | 36.8 | 27.4 |
| | Ground | 15 | 19.8 | 17.9 |

In addition to temperature, the green roof was more extreme in terms of solar radiation and substrate moisture (Table 3). Across the months of the growing season, solar radiation was higher on the green roof and substrate moisture content was lower. For both solar radiation and moisture content, the difference between the sites on the green roof and at grade was greatest in July and August.

**Table 3.** The average monthly solar radiation and substrate moisture content that were recorded on the green roof and at grade.

| Month | Site Location | Solar Radiation (W/m²) | Substrate Moisture (m³/m³) |
|-------|---------------|------------------------|----------------------------|
| May | Roof | 226.2 | 0.15 |
|  | Ground | 148.8 | 0.34 |
| June | Roof | 269.4 | 0.14 |
|  | Ground | 149.2 | 0.36 |
| July | Roof | 237 | 0.12 |
|  | Ground | 113.6 | 0.37 |
| August | Roof | 241.2 | 0.09 |
|  | Ground | 83.7 | 0.33 |
| September | Roof | 210.1 | 0.06 |
|  | Ground | 176.1 | 0.26 |

The results of the general linear mixed-effects regression model analysis showed that floral phenology was accelerated on the green roof for *Allium 'Millennium'* and *Ipomopsis aggregata* (Figures 1 and 2). *Allium 'Millennium'* green roof replicates had significantly more flowers than the replicates at grade starting at Week 9 ($p < 0.05$, Table 4). At Week 15, the *Allium 'Millennium'* grown at grade had higher average flower counts than those grown on the green roof ($p < 0.05$). *Ipomopsis aggregata* grown on the green roof had significantly more flowers beginning at Week 13 ($p < 0.05$, Table 5) until Week 17 ($p > 0.05$), when both the plants on the green roof and the plants at grade began to senesce. The *Oenothera speciosa* green roof replicates began blooming two weeks earlier than the replicates at grade, and the *Asclepias incarnata* green roof replicates began blooming four to five weeks earlier than the replicates at grade (Table 6). The accelerated floral phenology of the plants on the green roof is expected based on evidence provided by experimental warming studies, since the green roof environment was consistently warmer than the sites at grade [18].

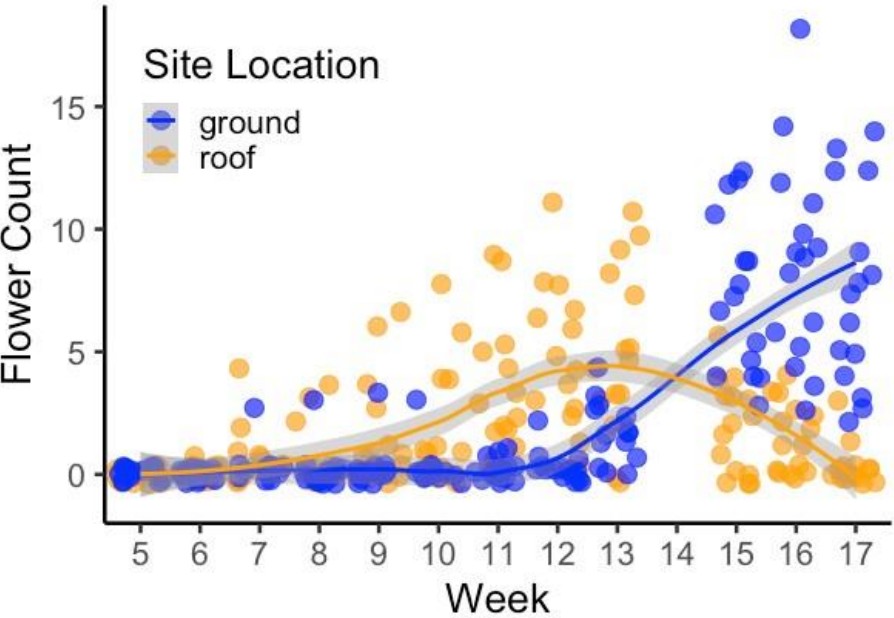

**Figure 1.** Floral phenology and abundance for *Allium 'Millennium'* beginning at Week 5 (5 June 2022) and ending at Week 17 (3 September 2022). A smooth curve was fitted using locally estimated parametric smoothing to visualize the relationship between plants on the ground and on the roof. Data were not collected in Week 14.

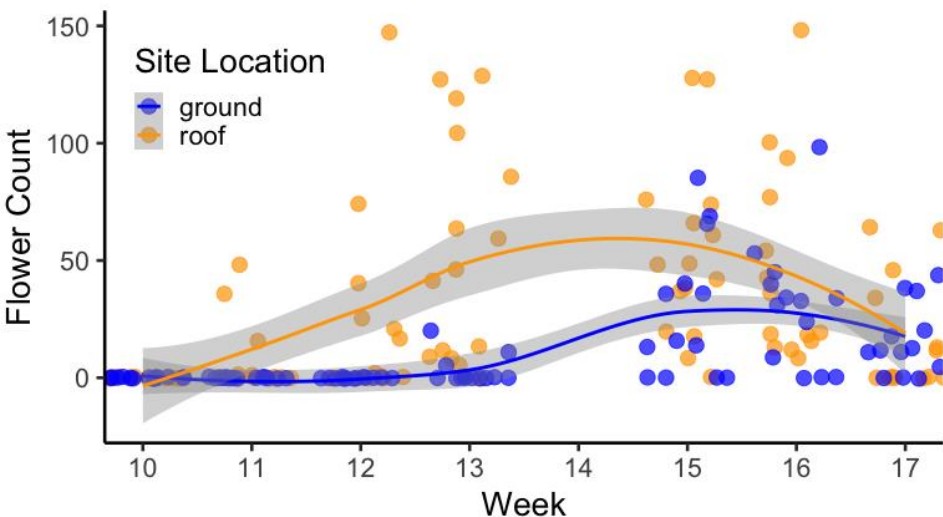

**Figure 2.** Floral phenology and flower abundance for *Ipomopsis aggregata* beginning at Week 10 (10 July 2022) and ending at Week 17 (3 September 2022). A smooth curve was fitted using locally estimated parametric smoothing to visualize the relationship between plants on the ground and on the roof. Data were not collected in Week 14.

**Table 4.** The standard error, z-scores, and *p*-values generated from the model depicting site-by-site comparisons of the floral phenology for *Allium 'Millennium'* at each week.

| Model Output for Ground—Roof Contrast of *Allium 'Millennium'* | | | |
|---|---|---|---|
| **Week Number** | **Standard Error** | **Z Score** | ***p*-Value** |
| Week 5 | 8917.93 | 0 | 1 |
| Week 6 | 6226.076 | −0.003 | 0.9978 |
| Week 7 | 0.686 | −1.535 | 0.1247 |
| Week 8 | 0.686 | −1.535 | 0.1247 |
| Week 9 | 0.633 | −3.214 | 0.0013 |
| Week 10 | 0.63 | −3.353 | 0.0008 |
| Week 11 | 0.616 | −4.426 | <0.0001 |
| Week 12 | 0.538 | −5.178 | <0.0001 |
| Week 13 | 0.276 | −3.671 | 0.0002 |
| Week 15 | 0.258 | 5.28 | <0.0001 |
| Week 16 | 0.283 | 6.558 | <0.0001 |
| Week 17 | 0.533 | 6.319 | <0.0001 |

**Table 5.** The standard error, z-scores, and *p*-values generated from the model depicting site-by-site comparisons of the floral phenology for *Ipomopsis aggregata* at each week.

| Model Output for Ground–Roof Contrast of *Ipomopsis aggregata* | | | |
|---|---|---|---|
| **Week Number** | **Standard Error** | **Z Score** | ***p*-Value** |
| Week 10 | 1879.91 | 0 | 0.9998 |
| Week 11 | 1273.01 | −0.015 | 0.9878 |
| Week 12 | 1273.49 | −0.016 | 0.9871 |
| Week 13 | 0.51 | −7.18 | <0.0001 |
| Week 15 | 0.484 | −2.638 | 0.0083 |
| Week 16 | 0.484 | −2.15 | 0.0316 |
| Week 17 | 0.49 | −1.438 | 0.1504 |

Table 6. The percentage of green roof replicates and replicates at grade in bloom each week for each species, beginning at Week 5 (5 June 2022) and ending at Week 17 (3 September 2022). Data were not collected in Week 14. GR represents the green roof and AG represents the ground.

| | | Percentage of Plant Replicates in Flower by Species and Site Location | | | | | | | |
|---|---|---|---|---|---|---|---|---|---|
| | *Genus Species* | *Allium 'Millennium'* | | *Oenothera speciosa* | | *Asclepias incarnata* | | *Ipomopsis aggregata* | |
| | **Site Location** | **GR** | **AG** | **GR** | **AG** | **GR** | **AG** | **GR** | **AG** |
| **Month** | **Week #** | | | | | | | | |
| June | 5 | 0 | 0 | 0 | 0 | 0 | 0 | 0 | 0 |
| | 6 | 7 | 0 | 0 | 0 | 0 | 0 | 0 | 0 |
| | 7 | 33 | 7 | 0 | 0 | 0 | 0 | 0 | 0 |
| | 8 | 20 | 7 | 0 | 0 | 0 | 0 | 0 | 0 |
| July | 9 | 53 | 7 | 13 | 0 | 0 | 0 | 0 | 0 |
| | 10 | 53 | 7 | 80 | 0 | 13 | 0 | 0 | 0 |
| | 11 | 87 | 20 | 53 | 33 | 40 | 0 | 33 | 0 |
| | 12 | 87 | 20 | 100 | 100 | 47 | 0 | 53 | 0 |
| August | 13 | 87 | 80 | 73 | 93 | 73 | 0 | 93 | 23 |
| | 14 | N/A | N/A | N/A | N/A | N/A | N/A | N/A | N/A |
| | 15 | 67 | 100 | 67 | 27 | 67 | 7 | 93 | 69 |
| | 16 | 60 | 100 | 67 | 7 | 67 | 7 | 93 | 77 |
| September | 17 | 13 | 100 | 13 | 13 | 13 | 13 | 47 | 77 |

Additionally, over 73% of the green roof replicates for *Asclepias incarnata* replicates bloomed during peak bloom, and 13% of the replicates at grade bloomed during peak bloom at Weeks 13 and 17, respectively. *Allium 'Millennium'* green roof replicates had the highest floral density between Weeks 11 and 13, and the replicates at grade had the highest floral density between Weeks 15 and 17. The *Allium 'Millennium'* green roof replicates reached peak bloom three weeks before the replicates at grade, and the peak bloom for the replicates on the green roof was less than the replicates at grade. The *Ipomopsis aggregata* green roof replicates reached peak bloom two or three weeks earlier than the replicates at grade, and the green roof replicates reached a higher flower count during peak bloom than the replicates at grade.

Earlier bloom times have potential synergies with urban pollinator conservation. Some bee species have been found to emerge earlier in the season, likely related to global temperature increases and the urban heat island effect [5]. Green roof plant species blooming earlier in the season than their ground-level counterparts may provide urban foraging opportunities that would otherwise not exist in their foraging area. An emerging phenomenon is being observed, the plant–pollinator phenological gap, which describes asynchronies forming between pollinators and their associated plant hosts. Linked to climate change and other warming effects, pollinators and plants emerge and bloom at altered times in the season, but not always at the same rates, causing a lack of pollinators for sexual plant reproduction or a lack of foraging opportunities for early emerging bees [19]. By leveraging the altered bloom times of green roof plants of affected species, we may be able to narrow the phenological gap between plants and some pollinators to better conserve and manage pollinator abundance and diversity in urban ecosystems. However, research on warming effects contributing to a phenological gap between plants and pollinators is limited. There is evidence that some pollinator species are negatively affected by accelerated floral phenology, as they emerge after their associated floral resources have finished blooming [20]. Thus, other interventions, in addition to green roofs, should be considered to address this gap to conserve pollinator diversity, and additional research on the effects warming has on the phenological gap between plants and pollinators is needed.

Plant size indices were calculated as a measure of relative plant health and cover at Week 6 and Week 19 (Figure 3). By Week 6 after planting, all three species *Allium*

'*Millennium*', *Asclepias incarnata*, and *Ipomopsis aggregata* were smaller at grade than on the green roof. However, by Week 19, all three species occupied a larger volume of space at grade than on the green roof. This same pattern is only reflected in the dry weights of *Oenothera speciosa*. This suggests that there are species-dependent effects on the plant growth form due to varied environmental factors between the green roof sites and the sites at grade. A likely explanation is the difference in light exposure between the two sites since abundant light is important immediately post-transplant and then factors such as heat and drought have impacts that become more prevalent later in the growing season. Additionally, there are likely differences in the nutrient-holding capacity and microbiota, which vary across soil textures, in addition to the recorded differences in moisture availability, which may have influenced the observed differences in plant growth and phenology between the sites on the green roof and the sites at grade.

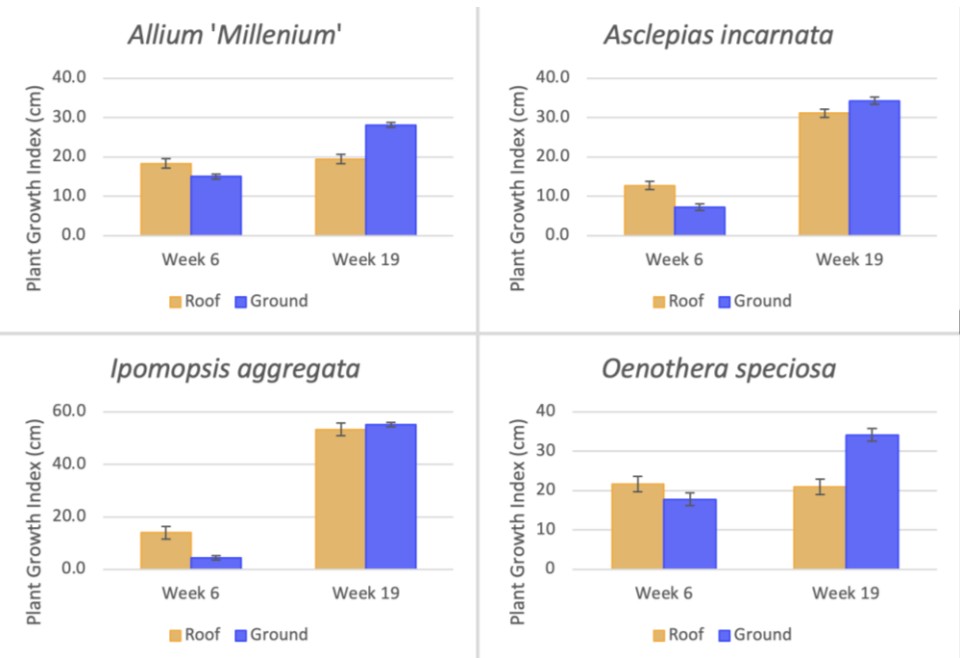

**Figure 3.** Plant sizes (cm) for *Allium 'Millennium'*, *Asclepias incarnata*, *Ipomopsis aggregata*, and *Oenothera speciosa* at grade and on the green roof at Weeks 6 and 19.

There were no significant differences in the average weight of the aboveground biomass of replicates grown on the green roof and replicates grown at grade for *Allium 'Millennium'*, Asclepias incarnata, and Oenothera speciosa (Figure 4). Ipomopsis aggregata replicates had more aboveground biomass on the green roof than the replicates at grade (Figure 5). There were significant differences between the dry, aboveground biomass for Ipomopsis aggregata green roof replicates and replicates at grade ($p < 0.001$). Though not statistically significant, the Oenothera speciosa replicates at grade generally had more aboveground biomass than the replicates at grade. These data suggest that the plant size response to green roof conditions is variable between plant species.

The relative chlorophyll content of *Allium 'Millennium'* was not significantly different between replicates grown on the green roof and replicates grown at grade (Figure 6). The average relative chlorophyll content of *Asclepias incarnata* and *Oenothera speciosa* was lower in replicates grown at grade than replicates grown on the green roof (Figure 6). Chlorophyll measurements were not made on *Ipomopsis aggregata* because of the shape of the leaf and limitations of the atLEAF chlorophyll meter.

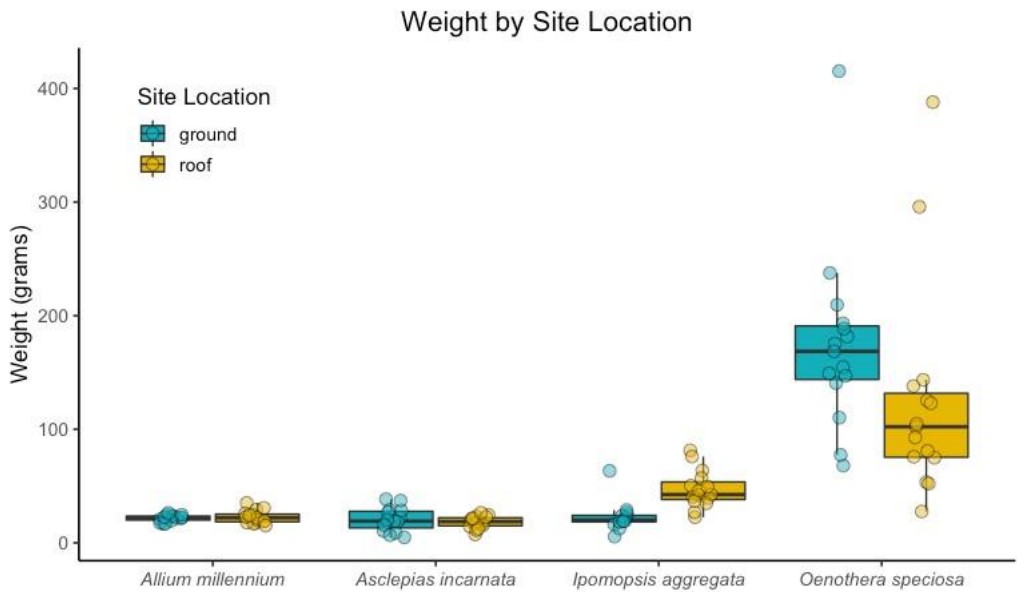

**Figure 4.** Dry weight (aboveground biomass) in grams, for each plant species on the green roof and at grade.

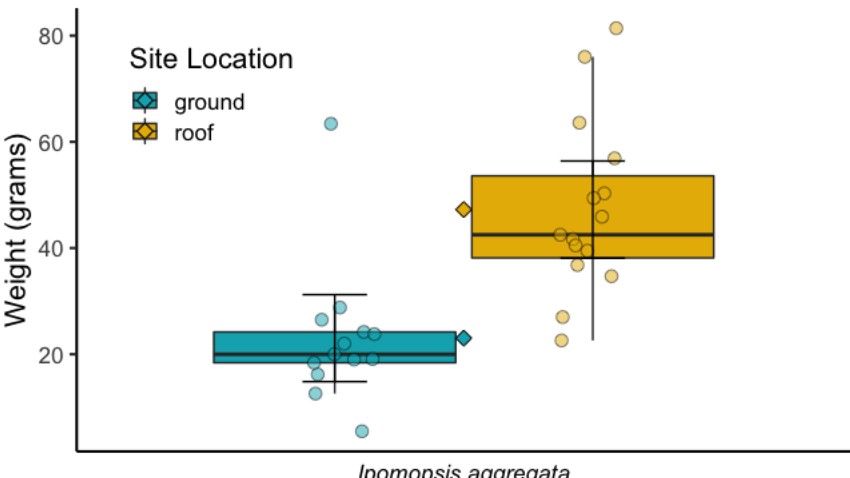

**Figure 5.** Dry weight (aboveground biomass) in grams, for *Ipomopsis aggregata* on the green roof and at grade. The diamonds represent the means for the green roof site and the site at grade.

Chlorophyll content is used as a measure of plant health and stress [21], which provides context for plant flowering. The relative chlorophyll content provides a metric for assessing plant health in the environment. For *Allium 'Millennium'* and *Oenothera speciosa*, there was no obvious difference and both species flowered profusely in the green roof plots and the plots at grade. However, *Asclepias incarnata* had a significantly lower relative chlorophyll content at grade than on the green roof, and most of the plants at grade did not bloom, despite there being no significant difference in the average weight of the aboveground biomass in the two locations. This suggests that *Asclepias incarnata* was able to generate more resources for floral production on the green roof than at grade in a single season.

Bee diversity and abundance were low for the in situ observations conducted at sites on the green roof and at grade (Table 7). No individuals belonging to the morphospecies group of the hairy leg bee were observed. Additionally, the traps did not collect any green metallic bees, cuckoo bees, or hairy belly bees (Table 8). Cuckoo bees and striped sweat bees were only observed on the green roof and bumblebees were only observed at grade. Green metallic bees were more commonly observed at grade than on the green roof. Striped

sweat bees were observed on the green roof, but not at grade. However, they were present in one of the transect traps and the trap on the bare roof. Tiny dark bees were only observed on the green roof but were found in both traps along the transect at grade. Hairy leg bees were not observed on the green roof or at grade but were present in all traps except for the trap on the green roof. Cuckoo bees were only observed on the green roof and were not present in any of the traps. Other groups including Coleoptera, Diptera, and Lepidoptera were observed in both locations and found in all traps except for the trap on the bare roof. The green roof trap had the most individuals belonging to the "other" category.

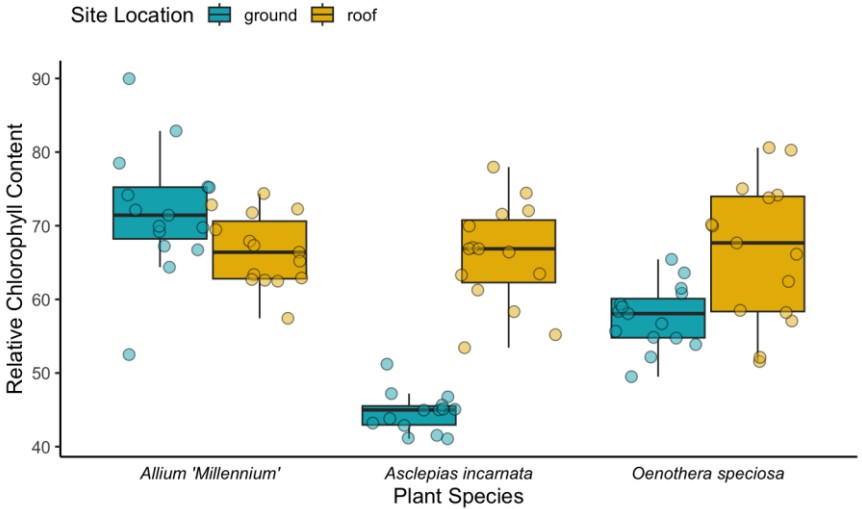

**Figure 6.** The relative chlorophyll content of each plant species grown on the green roof (GR) and at grade (AG). The teal boxplots represent replicates grown at grade and the yellow boxplots represent replicates grown on the green roof.

**Table 7.** The bee abundance and diversity recorded from in situ observations at the sites at grade (AG) and the sites on the green roof (GR).

| Honeybee | | Bumble Bee | | Green Metallic Bee | | Hairy Belly Bee | | Striped Sweat Bee | | Tiny Dark Bee | | Hairy Leg Bee | | Cuckoo Bee | | Other | |
|---|---|---|---|---|---|---|---|---|---|---|---|---|---|---|---|---|---|
| AG | GR | AG | GR | AG | GR | AG | GR | AG | GR | AG | GR | AG | GR | AG | GR | AG | GR |
| 4 | 7 | 1 | 0 | 8 | 1 | 0 | 1 | 0 | 3 | 0 | 1 | 0 | 0 | 0 | 9 | 5 | 24 |

**Table 8.** The bee diversity and abundance data, where (a) corresponds to one trap along a transect between the site at grade and the green roof site, (b) corresponds to the second trap along the transect, (c) corresponds to the trap on the non-green roof, (d) corresponds to the trap on the green roof, and (e) corresponds to the trap within the plots at grade.

| | Honeybee | Bumble Bee | Green Metallic Bee | Hairy Belly Bee | Striped Sweat Bee | Tiny Dark Bee | Hairy Leg Bee | Cuckoo Bee | Other |
|---|---|---|---|---|---|---|---|---|---|
| a. | 4 | 1 | 0 | 0 | 1 | 1 | 6 | 0 | 3 |
| b. | 2 | 0 | 0 | 0 | 0 | 1 | 1 | 0 | 1 |
| c. | 0 | 0 | 0 | 0 | 1 | 0 | 2 | 0 | 0 |
| d. | 2 | 0 | 0 | 0 | 1 | 0 | 0 | 0 | 7 |
| e. | 0 | 0 | 0 | 0 | 0 | 0 | 8 | 0 | 2 |

The methods used for pollinator observations may have contributed to the low abundance and diversity of bee observations. Anecdotally, bees were often observed on plants outside of the experimental observation window. Therefore, they were not counted in the data set. Additionally, only observing between 9:00 am and 11:00 am may have limited

what morphospecies we were able to observe. In a follow-up study, it would be worth incorporating an additional sample method and additional time frames for sampling to allow for a more complete sampling of the abundance and diversity of bees and other pollinators present at each location.

## 4. Conclusions

In conclusion, the green roof environment generally experienced higher temperatures, more drought, and higher light intensity. The four plant species that we were able to evaluate displayed accelerated floral phenology in the green roof environment. Plant health and growth metrics varied between species, suggesting some species perform better in green roof growing conditions while others perform better in traditional gardens. Pollinator abundance and diversity were similar between the green roof plots and the plots at grade. However, data on species diversity and abundance are not presented. With proper plant selection, green roofs can extend the flowering window when paired with analog gardens at grade. Additional research is needed to determine how generalizable these results are across plant species as there are few studies that directly compare the phenological differences of plants grown on a green roof compared to those at grade. The phenological gap between some plants and pollinators should be considered when considering green roofs for pollinator conservation as green roofs may further accelerate the floral phenology of plants upon which pollinators are reliant. The accelerated floral phenology may benefit some pollinator species and be of no use to others. In either case, a green roof provides a planted space that is otherwise unavailable with other roofing options and appears to offer expanded foraging opportunities for pollinators that are not typically offered synchronously in analogous garden settings at grade.

**Author Contributions:** Conceptualization, K.M.R. and J.M.B.; Methodology, K.M.R. and J.M.B.; Writing—original draft, K.M.R.; Writing—review & editing, K.M.R. and J.M.B. All authors have read and agreed to the published version of the manuscript.

**Funding:** This research received no external funding.

**Data Availability Statement:** Data is contained within the article.

**Acknowledgments:** Izzy ter Kuile, Jason Watson, and David Hansen from Colorado State University Facilities provided space and irrigation for the research sites. Lisa Mason provided training in the bee sampling protocol. High Plains Environmental Center and Arbor Valley supplied the plant material used in this study. Blake Gornick and Tatianna Hall, summer interns, provided support in the maintenance of the plots and data collection during the field season of 2022. Michael Guidi provided valuable guidance and feedback on the development of questions and methods posed by the authors.

**Conflicts of Interest:** The authors declare no conflicts of interest.

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
