# Peer review of "Green Roofs Affect the Floral Abundance and Phenology of Four Flowering Plant Species in the Western United States"

_land, doi:10.3390/land13010115_

Round 1

Reviewer 1 Report

Comments and Suggestions for Authors

The article is original and relevant. The problem of phenological discrepancies in the seasonal activity of plants and insect pollinators is becoming very serious in a warming climate. These inconsistencies, as the authors rightly point out, can lead to negative consequences not only for nature, but also for various branches of human activity. A very interesting approach has been chosen: the formation of in situ habitats with different microclimatic conditions. This allows us to solve the problem of phenological discontinuities to a certain extent. The methodological approach does not raise any questions. The article is factual, at the same time, the article notes some inattention when submitting materials. Thus, lines 212-213 conclude that "In every month other than July, the minimum temperatures on the green roof were less than those at grade (Table 2.)". However, in Table 2, the data is the opposite. And they are logical, since the minimum temperatures at altitude should be higher than at ground level.

Figure 3 shows the sizes of some plants after 6 and 19 weeks of growth. The difference in size (at 6 weeks the plants on the roof have larger sizes, after 19 weeks the plants on the ground) is explained by "A likely explanation is the difference in light exposure between the two sites" (lines 309-310). This assumption requires explanation, since illumination is a constant factor and it is obviously higher on the roof than on the ground. Therefore, the question remains why plants on ground suddenly became larger at week 19, although they were smaller at week 6.

There seems to be confusion in Table 7. It is called "The relative chlorophyll content of each plant species grown on the green roof (GR) and at grade (AG). The teal boxplots represent replicates grown at grade and the yellow boxplots 377 represent replicates grown on the green roof». However, the contents of the table do not correspond to the name at all.

It is necessary to analyze the text more carefully for such inconsistencies. It should also be noted that the Results and Discussion blocks are combined into one. If this is allowed by the editorial board of the journal, then at its discretion, but it is customary to separate them. However, there is no comparison of the results obtained with the results of similar studies, as is usually done in the Discussion section. There is also no Conclusion section, which is necessary in scientific articles.

Reviewer 2 Report

Comments and Suggestions for Authors

General Comments:

Interesting manuscript regarding the difference in phenological development of four plant species when comparing ground and roof-top growing conditions and the importance for pollinators. The manuscript is generally well-written and in good structure, however some clarifications are required, primarily in the methods, along with some additional discussion points.

The main concern of the manuscript is the skewing or perhaps omission of relevant information regarding the plant-pollinator relationship in view of climate change. While it is true that both plants and pollinators have been observed to change their phenological timing, the overwhelming majority is for plants to flower earlier and pollinators to remain stable, and thus lacking their usual resources. Therefore, even earlier flowering experienced on green-roofs is not likely to provide additional resources to most pollinators. This important point is mostly lacking in the introduction and discussion and must be substantially enhanced. Therefore, the authors must reflect on this apparent imbalance and provide specific examples from the literature on how this issue has been approached and discussed previously. This should be done both in the introduction, and in relation to the findings in the discussion.

Also, the title should be more specific, as currently it is too generic and broad for this particular study. Either include the location, or the specifics regarding the four species observed, preferably both.

Specific Comments:

1.     Lines 11-12. Some information regarding the pollinator observation method is required in the abstract.

2.     Lines 91-92. What is the structure and composition of the ‘native soils’? More information is required here.

3.     Lines 104-106. More information regarding the number of replicates for the different species is required.

4.     Lines 106-107. List the other species, as they can be interpreted to have affected the measurements and is therefore important to the reader and interpretation even if they did not flower.

5.     Line 110. Provide figure with design that showcases both the roof and at grade block designs with schematics, put in supplementary if required.

6.     Lines 110-115. How was the roof blocks 380 cm and beyond shallow, as it was stated earlier that the roof had a max depth of 60 cm?

7.     Lines 146-147. If no flower count data was collected for this species, how was bloom determined in lines 246-248 and Table 6? Clarification required.

8.     Lines 248-249. Where is the corresponding phenology/abundance flower count data for Aesclepias incarnata?

9.     Lines 308-310. Another very relevant factor is the soil and nutrients, this should be discussed as well.

Figures and Tables:

1.     Table 5. Header should probably be Ipomopsis aggregata, rather than Allium 'Millennium'.

2.     Figure 3. One axis is missing.

3.     Figure 6. The resolution of this figure is very poor in comparison to the others.

Round 2

Reviewer 1 Report

Comments and Suggestions for Authors Dear colleagues! Thank you for taking my comments into account. I am quite satisfied with your corrections and additions. The article is good and I wish you success!

Author Response

Thank you again for your time and insightful comments.

Reviewer 2 Report

Comments and Suggestions for Authors

General Comments:

The authors have sufficiently responded to the reviewers’ concerns and altered the manuscript accordingly.

Author Response

(The authors gave the same response as above.)
